precision medicine; therapeutics; treatment approaches; cancer; clinical trials; pancreatic cancer

**Authors for correspondence:**
Raffaella Casolino and Andrew V. Biankin,
Emails: raffaella.casolino@glasgow.ac.uk;
andrew.biankin@glasgow.ac.uk

# Treatment of pancreatic cancer in 2022

Raffaella Casolino[1] [iD] and Andrew V. Biankin[1,2,3]

[1]Wolfson Wohl Cancer Research Centre, Institute of Cancer Sciences, University of Glasgow, Glasgow, UK; [2]West of Scotland Pancreatic Unit, Glasgow Royal Infirmary, Glasgow, UK and [3]South Western Sydney Clinical School, Faculty of Medicine, University of NSW, Liverpool, NSW, Australia

## Abstract

Pancreatic ductal adenocarcinoma (PDAC) is usually diagnosed at an advanced, incurable, stage and has an extremely poor prognosis. Systemic chemotherapy represents the standard treatment either in the pre-operative, adjuvant and palliative setting, which is associated with only modest improvement in survival. More recently, advances in cancer genomic sequencing have unravelled the molecular heterogeneity of PDAC and identified small patient subgroups harbouring unique actionable aberrations in *BRCA*, *NTRK*, *NRG1* and mismatch repair genes paving the way to a more personalised approach for this tumour. However, the evolution of PDAC treatment towards a successful precision approach presents many challenges. In this review, we discuss the current standard treatments of PDAC, from early stage to advanced disease, and we illustrate the opportunities and challenges of precision medicine for this deadly cancer.

## Impact statement

This review discusses the current treatment strategies of pancreatic cancer in a comprehensive manner.

## Introduction

Pancreatic ductal adenocarcinoma (PDAC) is one of the most aggressive solid cancers, diagnosed at an advanced stage in most cases, whose treatment paradigm is based on chemotherapy with a few exceptions that are amenable to personalised therapeutic approaches (Landman et al., 2020; Huang et al., 2021). Despite major efforts in pre-clinical and clinical research and recent improvements in multimodality care, the mortality for patients with PDAC is one of the highest amongst solid tumours with an overall 5-year survival of less than 10% (Kamisawa et al., 2016; Gaddam et al., 2021; Rahib et al., 2021). Multi-omic studies from collaborative international networks have elucidated the molecular complexity of PDAC, which challenges the development of effective treatments (Biankin et al., 2012; Chang et al., 2014; Waddell et al., 2015; Cancer Genome Atlas Research, 2017; Connor et al., 2017; Aguirre et al., 2018; Qian et al., 2018; Singhi et al., 2019). Actionable molecular aberrations in *BRCA*, *NTRK*, *NRG1* and mismatch repair (MMR) genes – for which approved matched treatments are currently available – are described in less than 10% of patients (Biankin et al., 2012; Chantrill et al., 2015; Waddell et al., 2015; Bailey et al., 2016; Lowery et al., 2017). Many other potential therapeutic vulnerabilities occur at very low frequency (<1%) making the standard clinical trial model inadequate to test biomarker-based therapeutics in rare patient subgroup. On the other hand, the knowledge of the complex interaction between genetic events, epigenetic alterations and the tumour microenvironment, including the immune system in driving PDAC initiation, progression and therapeutic response is still limited. As a result, current standard treatment is largely based on unselected approaches and progress in therapeutic development lags behind other tumour types (Swanton et al., 2016). This review discusses the treatment of PDAC in 2022 and novel promising therapeutic approaches under clinical investigation.

## Treatment of early-stage PDAC

Upfront surgery followed by adjuvant treatment is considered the treatment of choice for patients with resectable PDAC and represents the only chance of cure (Tempero et al., 2021). Due to late clinical presentation, high metastatic potential and lack of effective screening strategies, less than 20% of patients are diagnosed with anatomically resectable tumours (Gillen et al., 2010). Improvements in surgical techniques and postoperative management have substantially enhanced local control and survival for patients with early stage PDAC over the last decades. However, the risk of recurrence after surgery is as high as 85% and the 5-year survival less than 30% (Oettle et al., 2013; Conroy et al., 2018; Groot et al., 2018; Strobel et al., 2019). The allocation of adjuvant chemotherapy has demonstrated to improve survival outcomes in several clinical

trials of resected PDAC patients. Neoadjuvant approaches are now coming to the fore with several trials reporting some benefit.

### Adjuvant treatment

The phase III ESPAC-1 and CONKO trials showed improved overall survival (OS) with adjuvant chemotherapy in patients with resected PDAC using 5-fluorouracil (5-FU modulated by leucovorin) and gemcitabine monotherapy, respectively (Neoptolemos et al., 2001; Oettle et al., 2007). No differences between the two agents in terms of efficacy have been identified by a subsequent study, the ESPAC-3 trial (Neoptolemos et al., 2012). More recently, combinatorial regimens have been tested changing the therapeutic paradigm for the adjuvant treatment of PDAC. The phase III ESPAC-4 trial demonstrated superior efficacy of the combination of gemcitabine (1,000 mg/m$^2$ once a week for 3 of every 4 weeks) and capecitabine (1,660 mg/m$^2$ daily for 21 days followed by 7 days break) over gemcitabine monotherapy with a median OS of 28.0 versus 25.5 months, respectively (HR 0.82; 95% CI, 0.68–0.98, $p$ = 0.032; Neoptolemos et al., 2017). A total of 732 patients with European Cooperative Oncology Group (ECOG) performance status (PS) 0–2 were randomised, regardless of serum CA19.9 levels. Sixty percent of them had R1 resection margin status and 80% had node-positive disease on histopathology. Six cycles of treatment were completed by 54% of patients in the investigational arm and 65% of patients in the gemcitabine arm. The 5-year survival rate was 28% for the combination and 20% for single-agent gemcitabine. Seventeen percent of patients had high post-operative CA19–9 serum levels, a known independent prognostic factor, suggesting the presence of early recurrence that might explain the superiority of the combination therapy over gemcitabine monotherapy for patients with CA19–9 > 92.5 IU/m (Neoptolemos et al., 2017).

In 2018, the PRODIGE24 phase III trial of 493 patients with resected PDAC randomised to 24 weeks of modified 5-fluorouracil (5 FU) plus leucovorin, oxaliplatin and irinotecan (with the 5 FU bolus omitted and dose reduced irinotecan, i.e., m-FOLFIRINOX) versus gemcitabine, showed impressive disease-free survival (DFS) of 21.6 versus 12.8 months in the investigational and standard group, respectively (HR 0.58; 95% CI, 0.46–0.73, $p$ < 0.001). In addition, OS was significantly improved in the m-FOLFIRINOX arm, with a median of 54.4 versus 35.0 months for gemcitabine (HR 0.64; 95% CI, 0.48–0.86; $p$ = 0.003). Median age of patients was 63 years; R1 resection rate was >40% and 75% had node-positive disease. Differently from the ESPAC-4, in the PRODIGE24 trial, the patient population was highly selected. Only patients with good clinical conditions (PS 0–1), low CA 19.9 serum levels (<180 U/mL) and normal postoperative CT scan were included, and centralised review of surgical and pathology reports was assessed. This trial has likely included patients with less risk of early relapse compared to the ESPAC-4 study. Sixty-six percent of patients in the m-FOLFIRINOX arm and 79% in the gemcitabine arm completed the course of chemotherapy. Grade 3 or 4 adverse events including diarrhoea, peripheral neuropathy, fatigue, nausea and vomiting were significantly higher in m-FOLFIRINOX group compared with the control arm (75.9 vs. 52.9%, respectively). Rates of neutropenia and febrile neutropenia were similar between the two groups; however, 62.2% of the m-FOLFIRINOX group required G-CSF versus only 3.7% of the control arm (Conroy et al., 2018).

Summarising, based on the available evidence, the standard of care for patients after radical resection of their PDAC should be combined chemotherapy. Although there is no direct comparison between gemcitabine-capecitabine and m-FOLFIRINOX, the triplet is thought to be the best treatment option for fit patients. The European Society for Medical Oncology (ESMO) guidelines recommend m-FOLFIRINOX as the first adjuvant therapeutic option after resection of PDAC in selected fit patients, in view of survival outcomes and associated toxicity profile. In more frail patients (age >70, ECOG PS 2, or patients who have any contraindication to FOLFIRINOX), gemcitabine-capecitabine could be an option. Gemcitabine alone should be used only in frail patients (Ducreux et al., 2015).

### Pre-operative systemic treatment

If on one hand adjuvant chemotherapy represents a key element of the multimodal management of patient with early stage PDAC, on the other up to 82% of patients are not able to receive any treatment after resection mostly due to postoperative morbidity (Mayo et al., 2012; Merkow et al., 2014). Besides, only 55–75% of those who initiate adjuvant therapy complete the treatment, where fully completed adjuvant chemotherapy is an independent prognostic factor for survival (Valle et al., 2014). When considering trials including patients with good PS and optimal recovery after surgery, only 66.4% of m-FOLFIRINOX patients in the PRODIGE study and 54% of patients of capecitabine-gemcitabine group in the ESPAC-4 trial received all the planned cycles. In this context, primary systemic treatment has been increasingly explored and adopted in patients with non-metastatic PDAC. This strategy consists of the allocation of systemic therapy in pre-operative setting and is based on the following rationale: in vivo chemosensitivity test, early treatment of occult micrometastases, decreased nodal involvement, increased radical resection rates, better selection of patients who are more likely to benefit from surgery, improved compliance with chemotherapy, and improved survival after curative resection (Laurence et al., 2011; Roland et al., 2015; Mokdad et al., 2017; Chawla and Ferrone, 2019). Pre-operative treatment was initially tested in borderline resectable and locally advanced PDAC with induction intent (Jang et al., 2018; Maggino et al., 2019). More recently, the investigation of this approach has been extended to patients with resectable disease as a pure neoadjuvant (NAT) strategy. However, data on its utility are still controversial with conflicting results on survival benefit compared with upfront surgery followed by adjuvant treatment in this population (Gillen et al., 2010; Xu et al., 2014; Bradley and Van Der Meer, 2019; Lee et al., 2019). NAT followed by resection was associated with improved OS compared with upfront resection in a large retrospective study with median OS of 26 versus 21 months, respectively (HR 0.72; 95% CI, 0.68–0.78). Patients who underwent upfront resection had higher T stage, positive lymph nodes, and R1 resection at histopathology examination. When compared with a subset of patients who received adjuvant therapy following resection, the NAT group had improved OS (HR 0.83; 95% CI, 0.73–0.89) (Mokdad et al., 2017). Two systematic reviews and meta-analyses did not identify survival benefit of NAT versus up-front surgery (HR 0.96; 95% CI, 0.82–1.12 (Lee et al., 2019) and HR 0.86; 95% CI, 0.73–1.03 (Ye et al., 2020) despite significant improvement in radical resection rate and the reduction of lymph nodes involvement. Only patients who completed NAT with subsequent resection had significantly increased OS versus surgery followed by adjuvant treatment in one study (HR 0.82; 95% CI, 0.71–0.93; Lee et al., 2019).

Perioperative strategy has been evaluated in SWOG1505, a phase II trial of resectable PDAC. A total of 102 patients were randomised to m-FOLFIRINOX or gemcitabine/nab-paclitaxel for

3 months preoperatively and 3 months postoperatively. Eighty-five percent of patients completed neoadjuvant therapy, 70% underwent resection and 55% were able to start adjuvant chemotherapy. R0 resection was 85% of patients in both arms and pathologic complete was achieved in 25 and 42% of cases, respectively. Overall, 49% in the m-FOLFIRINOX arm and 40% in gemcitabine/nab-paclitaxel arm completed all treatment. The primary endpoint of 2-year OS was 47% for m-FOLFIRINOX and 48% for gemcitabine/nab-paclitaxel, while median OS 23.2 and 23.6 months, respectively, substantially similar to historical data of patients undergoing surgery followed by adjuvant chemotherapy.

Summarising, given the lack of robust evidence from randomised clinical trials, to date NAT should be offered only in the context of clinical studies to patients with resectable disease (Table 1). Only those at risk of developing postoperative complications (particularly pancreatic fistula) or those with high-risk characteristics (i.e., suspicious of advanced disease based on clinical, radiological or serum findings) may be eligible for NAT after multidisciplinary discussion (Khorana et al., 2019; Pentheroudakis and Committee, 2019; Tempero et al., 2019).

In patients with borderline resectable tumour, neoadjuvant therapy is increasingly adopted as the preferred therapeutic choice to up-front surgery, although prospective randomised evidence supporting this strategy is limited for patients with this tumour stage (Table 1). The first large phase III study of neoadjuvant chemoradiotherapy, the PREOPANC trial conducted in the Netherlands, randomised 246 patients with resectable and borderline resectable PDAC to either upfront surgery followed by adjuvant therapy, or neoadjuvant treatment, surgery and adjuvant gemcitabine. Neoadjuvant chemoradiotherapy consisted of three cycles of gemcitabine combined with 36 Gy radiotherapy in 15 fractions during the second cycle. After restaging, patients underwent surgery followed by four cycles of adjuvant gemcitabine. Patients in the upfront surgery group underwent surgery followed by six cycles of adjuvant gemcitabine. Patients with resectable and borderline resectable tumours were 55 and 45%, respectively. The neoadjuvant chemoradiotherapy arm who underwent surgery had higher rate of R0 resection (72 vs. 43%) and higher rate of node-negative resections (65 vs. 18%). In the intention to treat analysis, OS was marginally improved in the neoadjuvant chemoradiotherapy group, with median OS of 15.7 versus 14.3 months (HR 0.73; 95% CI, 0.56–0.96, $p = 0.025$). Median OS was 33.7 months with neoadjuvant chemoradiotherapy versus 17.3 months with upfront surgery (HR 0.47, 95% CI 0.32–0.67, $p < 0.01$); the 5 year-survival rates were 33.9 versus 8.4%, respectively. The effect of neoadjuvant chemoradiotherapy was consistent across the prespecified subgroups, including resectable and borderline resectable PDAC (Versteijne et al., 2022).

In locally advanced unresectable PDAC (LAPC) primary systemic therapy constitutes the standard of care (Table 1). Patients with this tumour stage have high rates of non-radical resection and outcomes are similar to patients who do not proceed to resection (Seufferlein et al., 2019). Conversion surgery can be considered as a potential option after multidisciplinary discussion and proposed in highly selected cases with optimal response after induction treatment. Gemenetzis et al. analysed 461 patients with LAPC defined according to National Comprehensive Cancer Network (NCCN) guidelines. Chemotherapy regimens included FOLFIRINOX (50%), gemcitabine-based (31%) or a combination (19%). Twenty percent underwent resection and 89% of these had radical resection, with median OS of 35.3 months for the resected group and 16.2 months for the non-resected group

(Gemenetzis et al., 2019). Findings from a meta-analysis on patients with LAPC who underwent surgical resection after induction FOLFIRINOX, show up to 43% of conversion surgery rate with a pooled percentage of 26% and radical resection rate ranging between 50 and 100% (Suker et al., 2016). In selected cases adding loco-regional therapy such as sequential chemoradiation or stereotactic body radiation therapy can be considered for local control in patients with LAPC following induction chemotherapy, however, due to conflicting results from published studies, this approach is still debatable (Table 1; Kleeff et al., 2019; Jung et al., 2019). Results from ongoing clinical trials are warranted to clarify if there is any benefit from this multimodal strategy.

Guidelines on pre-operative treatment in non-metastatic PDAC, which are summarised in Table 1, are based on systematic reviews of cohort studies given the lack of phase III trials in this setting (Oxford Levels of Evidence category 2A) (Ducreux et al., 2015; Khorana et al., 2019; Pentheroudakis and Committee, 2019; Tempero et al., 2019). In addition, the heterogeneous design of studies investigating the role of pre-operative systemic treatment in PDAC (i.e., prospective, randomised trials and meta-analysis evaluating single agents, polychemotherapy or chemoradiation, with different inclusion criteria) and the lack of consensus regarding the definition of what precisely constitutes resectable, borderline resectable and locally advanced – unresectable disease, make interpretation of study results challenging with uncertainty about the optimal chemotherapy regimen or its optimal timing and duration (Ferrone et al., 2015; Gilbert et al., 2017; Heinrich et al., 2019; Janssen et al., 2020; Kulkarni et al., 2020; Oba et al., 2020; Ye et al., 2020).

By extrapolating data from the metastatic setting, current guidelines consider FOLFIRINOX, m-FOLFIRINOX, gemcitabine or gemcitabine/nab-paclitaxel an acceptable option in the neoadjuvant/borderline resectable setting or in locally advanced tumours (category 2A) (Ducreux et al., 2015; Khorana et al., 2019; Pentheroudakis and Committee, 2019; Tempero et al., 2019, 2021). Multimodal treatment with chemoradiotherapy can be considered in selected cases, but the conclusions about its efficacy are controversial (Jang et al., 2018; Reni et al., 2018; Kleeff et al., 2019; Pan et al., 2019; Ghaneh et al., 2020; Versteijne et al., 2020). Additional strategies such as perioperative treatments showed early promising results but need further investigation (Sohal et al., 2020). Prospective randomised trials are needed to establish the optimal regimen and to demonstrate the potential benefits of pre-operative therapeutic approaches. Implementation of biomarker-based clinical trials will be critical to tailor the therapeutic strategy on tumour molecular profile, which is currently unmet need in this setting (Casolino et al., 2020). Lastly, referring patients with non-metastatic tumour to high-volume centres is strongly recommended as well as enrolment in clinical trials given the limited robust evidence on preopearative regimens off-study (Tempero et al., 2019).

## Standard treatment of metastatic PDAC

### Unselected approach: First-line

Metastatic disease accounts for over 50% of patients. In this setting, the intent of treatment is palliative, and chemotherapy still represents the standard of care. The possibility of surgical treatment for metastatic disease can be considered in highly selected patients, however, evidence supporting this approach is limited and derived from retrospective data (Antoniou et al., 2016; Yu et al., 2017).

**Table 1.** PDAC resectability status and associated treatments

| Resectability status[a] | Standard treatment[b] |
|---|---|
| **Resectable**<br>• No arterial tumour contact (CA, SMA, CHA)<br>• No tumour contact with the SMV or PV or ≤180° contact without vein contour irregularity | Surgery followed by adjuvant treatment<br><br>Consider staging laparoscopy and neoadjuvant therapy, particularly in high-risk patients[c] |
| **Borderline resectable**<br>*Pancreatic head/uncinate process:*<br>• Solid tumour contact with CHA without extension to CA or hepatic artery bifurcation allowing for safe and complete resection and reconstruction<br>• Solid tumour contact with the SMA of ≤180°<br>• Solid tumour contact with variant arterial anatomy and the presence and degree of tumour contact should be noted if present, as it may affect surgical planning<br>*Pancreatic body/tail:*<br>• Solid tumour contact with the CA of ≤180°<br>• Solid tumour contact with the CA of >180° without involvement of the aorta and with intact and uninvolved gastroduodenal artery thereby permitting a modified Appleby procedure (some panel members prefer these criteria to be in the locally advanced category)<br>• Solid tumour contact with the SMV or PV of >180°, contact of ≤180° with contour irregularity of the vein or thrombosis of the vein but with suitable vessel proximal and distal to the site of involvement allowing for safe and complete resection and vein reconstruction<br>• Solid tumour contact with the inferior vena cava (IVC) | Neoadjuvant therapy followed by surgery<br>Consider staging laparoscopy |
| **Locally advanced**<br>Unreconstructible SMV/PV due to tumour involvement or occlusion (can be due to tumour or bland thrombus)<br>*Head/uncinate process:*<br>• Solid tumour contact with SMA >180°<br>• Solid tumour contact with the CA >180°<br>*Pancreatic body/tail:*<br>• Solid tumour contact of >180° with the SMA or CA<br>• Solid tumour contact with the CA and aortic involvement | Clinical trial (preferred)<br><br>Induction chemotherapy (preferably 4–6 months) followed by chemoradiation or stereotactic body RT (SBRT) in selected patients (locally advanced without systemic metastases) or chemoradiation, or SBRT in selected patients who are not candidates for combination therapy |

*Note*: National Comprehensive Cancer Network (NCCN) definition of resectability, based on the anatomic contact on imaging of tumour and blood vessel.

Abbreviations: AO, aorta; BR, borderline resectable; CA, celiac axis; CHA, common hepatic artery; IVC, inferior vena cava; LA, locally advanced unresectable; PHA, proper hepatic artery; PV, portal vein; R, resectable; SMA, superior mesenteric artery; SMV, superior mesenteric vein.

[a]Decisions about resectability status should be made in consensus at multidisciplinary discussions.

[b]Participation in clinical trials should be preferred.

[c]High-risk patients: CA 19–9 more than 500 IU/ml, regional lymph node metastasis (biopsy or PET-CT), poor performance status (Eastern Cooperative Oncology Group score = 2, or more).

Gemcitabine and fluoropyrimidine-based combination regimens have showed efficacy in patients with advanced PDAC in phase III clinical trials, although median OS is approximately 1 year with the best combinatorial regimens. The two most effective options are FOLFIRINOX and gemcitabine/nab-paclitaxel. In the ACCORD11/PRODIGE4 phase III trial of 342 patients with untreated advanced PDAC, FOLFIRINOX was superior to gemcitabine monotherapy with regards to OS, progression-free survival (PFS) and objective response rate (RR). The median OS was 11.1 versus 6.8 months (HR 0.57; 95% CI, 0.45–0.73, $p < 0.001$), PFS 6.4 versus 3.3 months (HR 0.47; 95% CI, 0.37–0.59, $p < 0.001$) and RR 31.6 versus 9.4%. Patients in the FOLFIRINOX arm had significantly more side effects, including grade 3 or 4 neutropenia (45.7 vs. 21%), diarrhoea (12.7 vs. 1.8%) and neuropathy (9 vs. 0%; Conroy et al., 2011).

The combination of gemcitabine/nab-paclitaxel has been established as another standard first-line regimen for PDAC in the MPACT phase III trial. This study of 861 patients demonstrated the superiority of gemcitabine/nab-paclitaxel versus gemcitabine alone with median OS of 8.5 versus 6.7 months (HR 0.72; 95% CI, 0.62–0.83, $p < 0.001$), PFS of 5.5 versus 3.7 months (HR 0.69; 95% CI, 0.58–0.82, $p < 0.001$) and RR of 23 versus 7%. The 1-year survival rate in the gemcitabine/nab-paclitaxel group was 35 versus 22% in the gemcitabine monotherapy group. The most common adverse events of grade 3 or higher were worse in the experimental

arm, including neutropenia (38 vs. 27%), febrile neutropenia (3 vs. 1%), fatigue (17 vs. 7%) and neuropathy (17 vs. 1%; Von Hoff et al., 2013).

Currently, no predictive biomarkers are available to select patient for the most appropriate treatment in first-line setting and the two regimens have not been compared in head to head trial. The two studies enrolled different population, with higher proportions of elderly patients, tumour burden, ECOG PS 2 in the MPACT trial making it impossible to directly compare outcomes. In a real-world retrospective data, the efficacy between the two regimens resulted similar in terms of OS and PFS (Pusceddu et al., 2019). Therefore, the choice between FOLFIRINOX and gemcitabine/nab-paclitaxel is currently guided by clinical parameters (patient age, comorbidities, vascular access), physician preference, local guidelines and reimbursement status.

Summarising, the ESMO and NCCN guidelines recommend patients with metastatic PDAC and ECOG PS 0 or 1 to receive first-line treatment with gemcitabine/nab-paclitaxel or FOLFIRINOX. Patients with ECOG PS 2 should generally receive gemcitabine monotherapy, or best supportive care in case of major comorbidities. In selected cases, gemcitabine/nab-paclitaxel can be considered a reasonable option if the poor PS is due to a heavy tumour load (Ducreux et al., 2015; Tempero et al., 2021). It is always advisable to enrol patients in clinical trial when possible.

## Unselected approach: Second-line

Due to rapid clinical deterioration, less than 50% of patients are able to receive a second-line treatment after progression on first-line (Tempero et al., 2021). Until recently there has been no approved standard of care for second-line treatment in PDAC and the choice after FOLFIRINOX commonly comprised gemcitabine/nab-paclitaxel (and vice versa) although without randomised evidence. Oxaliplatin-based regimens have been investigated in the CONKO-003 and PANCREOX studies, which yielded conflicting results. The CONKO-003 phase III study reported a minimal survival benefit with second-line fluorouracil (FU) and oxaliplatin using the oxaliplatin, folinic acid and FU (OFF) regimen (Oettle et al., 2014). The median OS in the OFF arm was 5.9 and 3.3 months in the 5-FU/leucovorin arm (HR 0.66; 95% CI, 0.48–0.91; $p = 0.01$). However, results from the phase III PANCREOX trial show that the addition of oxaliplatin to 5-FU/leucovorin in subsequent treatment may be detrimental (Gill et al., 2016). More recently, in the phase III NAPOLI-1 study conducted in patients with metastatic PDAC previously treated with gemcitabine, combination of nanoliposomal irinotecan with 5-fluorouracil (5-FU) and folinic acid leucovorin (LV) has shown superior OS (6.1 vs. 4.2 months), PFS and RR in the intent-to-treat population over 5-FU/LV alone (Wang-Gillam et al., 2016).

Guidelines recommend second-line therapy to be considered in terms of risk-benefit for the patient. With a manageable safety profile, for fit patients refractory to first-line gemcitabine-based therapy, nanoliposomal irinotecan combined with 5-FU and LV may constitute an active and tolerable second-line treatment option. Other therapeutic options, based on less robust evidence, may be considered and include FOLFIRI, FOLFIRINOX or modified FOLFIRINOX, FOLFOX, capecitabine-oxaliplatin, capecitabine or continuous infusion 5-FU (Ducreux et al., 2015; Tempero et al., 2021). Given the limited impact on survival with the available therapeutic options in second-line setting, it is advisable to consider patients for inclusion in clinical trials.

To summarise, second-line treatment options for patients with PDAC previously treated with gemcitabine-based therapy and good PS include: 5-FU/leucovorin/liposomal irinotecan (category 1 for metastatic disease according to NCCN guidelines), FOLFIRI, FOLFIRINOX or modified FOLFIRINOX, 5-FU/leucovorin/oxaliplatin (OFF), FOLFOX, capecitabine-oxaliplatin, capecitabine or continuous infusion 5-FU. Options for patients with good PS and previously treated with fluoropyrimidine-based therapy include: 5-FU/leucovorin/nanoliposomal irinotecan (if no prior irinotecan administered), gemcitabine/nab- paclitaxel, gemcitabine/cisplatin or gemcitabine monotherapy. Treatment options for patients with poor PS include gemcitabine, capecitabine and continuous infusion 5-FU (Ducreux et al., 2015; Tempero et al., 2021). Therapeutic options for patients with actionable molecular alterations are discussed in the next paragraph.

## Precision medicine approaches

Trials investigating targeted agents have been largely unsuccessful for patients with PDAC when based on unselected approach. More recently, advances in cancer sequencing have unravelled the molecular heterogeneity of PDAC and identified small patient subgroups harbouring unique actionable aberrations in *BRCA*, *NTRK*, *NRG1* and MMR genes which paved the way to personalised medicine for this tumour. Despite the low frequency of most individual genetic alterations, 25–40% of PDAC patients harbour

at least one actionable molecular event that can potentially be therapeutically targeted with biomarker-matched therapies. However, definitive data about the clinical benefit of this strategy are limited (Aguirre et al., 2018; Pishvaian et al., 2020).

Patients with metastatic PDAC and deleterious or suspected deleterious germline *BRCA* mutation whose disease has not progressed after at least 16 weeks of first-line platinum-based chemotherapy, can potentially benefit from olaparib given as maintenance therapy (Golan et al., 2019). In the phase III POLO study conducted in this population, maintenance therapy with the PARPi olaparib significantly improved PFS versus placebo (7.4 vs. 3.8 months, $p = 0.004$) with RR of 23 versus 12%, respectively, although no significant difference in OS where observed (Golan et al., 2019). These results led to approval of olaparib in multiple countries as maintenance therapy after platinum-based first-line in patients with advanced PDAC associated with germline *BRCA* mutation.

In recent years, the U.S. Food and Drug Administration (FDA) has granted approval to other several compounds, based on tumour-agnostic basket trials, which can also be available for patients with PDAC including: the immune checkpoint inhibitor (ICI) pembrolizumab in tumours with microsatellite instability (MSI) (Le et al., 2015, 2017); the tropomyosin receptor kinase (TRK) inhibitors larotrectinib or entrectinib in tumours with neurotrophic tyrosine/tropomyosin receptor kinase (NTRK) gene fusion (Drilon et al., 2018; Doebele et al., 2020) and zenocutuzumab for patients with ligand neuregulin (NRG1) gene fusions (Schram et al., 2019; Table 2).

Despite occurring in 1–2% of patients with PDAC, germline mutations in the MMR genes MLH1, MSH2, MSH6 or PMS2 associated to MSI have important therapeutic implications in this tumour. Pembrolizumab, an ICI which blocks the interaction between programmed cell death-1 (PD-1) receptor and its ligands PD-L1 and PD-L2, has been tested in an early phase study of MSI-High cancers. Of eight patients with PDAC enrolled in the study, three (37%) had partial response (PR) and two achieved complete response (CR) (25%) (Le et al., 2017). The KEYNOTE-158 trial showed that of 22 patients with MSI-High PDAC, one had CR and three PR, corresponding to an overall RR of 18.2% (95% CI; 5.2–40.3). Median PFS was 2.1 months (95% CI; 1.9–3.4), median OS was 4.0 months (95% CI; 2.1–9.8), and median duration of response was 13.4 months. The group of non-colorectal MSI-High cancer achieved 34% of RR (95% CI; 28.3–40.8) and median OS of 23.5 months (95% CI; 13.5–not reached) (Marabelle et al., 2020). Based on these findings, pembrolizumab has granted approval by the FDA for treatment of refractory MSI-High cancers regardless tumour site, including PDAC. More recently pembrolizumab has also been approved by the FDA for cancers with a tumour mutation

**Table 2.** Precision therapeutic opportunities in PDAC

| FDA-approved precision-based therapies potentially available in PDAC |
| --- |
| **BRCA mutations: olaparib**<br>*NTRK* fusions: Larotrectinib[a], entrectinib[a]<br>*MSI-H* status: Pembrolizumab[a]<br>*BRAF* mutations: Encorafenib/binimentinb<br>*ROS1* fusions: Entrectinib<br>*ALK* fusions: Crizontinib, ceritinib, alectinib<br>*RET* fusions: Pralsetinib<br>*NRG1* fusions: Afatinib |

*Note:* Therapeutic strategies based on the presence of driver aberrations approved in PDAC (bold) or in other cancer types, potentially active in PDAC (roman).
[a]FDA-approved molecules with tissue-agnostic indications against the noted molecular alteration.

burden (TMB) ≥10 mutations per megabase (Mb), based on the KEYNOTE-158 clinical trial showing 29% of RR (95% CI; 21.0–39.0) in TMB–high cancers, compared with 6% (95% CI; 5.0–8.0) in non–TMB–high cancers (Subbiah et al., 2020). However, patients with PDAC were not included in the study.

NTRK1/2/3 fusions are predictive biomarkers of therapeutic response to NTRK inhibitors entrectinib and larotrectinib, which are FDA-approved in a tumour-agnostic fashion for the treatment of patients with these molecular alterations (Dunn, 2020). Notwithstanding rare frequency of NTRK1/2/3 fusions in PDAC (1%), encouraging results from case series showing long-term benefit of TRK inhibitors in TRK-fusion PDAC, support early testing for these gene fusions to allow timely identification of patients for targeted therapy (Pishvaian et al., 2018; Gupta et al., 2021).

*NRG1* gene fusions are extremely low in PDAC (<0.5%) although their frequency seems to be enriched in *KRAS*-wild-type tumours (Nagasaka and Ou, 2022). In 2021, the FDA has granted a Fast Track designation to zenocutuzumab (MCLA-128) as a potential treatment option for patients with metastatic solid tumours harbouring NRG1 gene fusions that have progressed on standard of care therapy. The global, open-label, multicenter phase II basket trial of zenocutuzumab (NCT02912949) is in progress with investigator-assessed RR as its primary end point. Patients in the study are divided into three cohorts, with cohort 2 enrolling NRG1 fusion-positive PDAC patients. Based on current evidence, it is advisable to screen for *NRG1* gene fusions allKRAS-wild type or early onset patients (Tempero et al., 2021).

With the entrance of multiple KRAS G12C inhibitors in clinical trials, screening for KRAS mutations will likely increase in the near future. In the phase I/II CodeBreaK100 study, Sotorasib, that is, an irreversible inhibitor of KRAS G12C approved by the FDA for the treatment of patients with non–small cell lung cancer and KRAS G12C mutations, showed promising activity in heavily pre-treated patients with metastatic PDAC and a KRAS G21C mutation (occurring in 1–2% of PDAC patients). This agent was associated with RR of 21.1% and disease control rate of 84.3% in 38 patients with PDAC enrolled in the study, achieving median PFS of 4 months and median OS of 6.9 months. Sotorasib was well tolerated, with manageable adverse events (Sotorasib Tackles KRASG12C-Mutated Pancreatic Cancer, 2022).

Last, several novel compounds targeting single gene alterations, immune modulation, metabolism and protein tropism are currently under early clinical investigation in metastatic PDAC and will hopefully soon expand the therapeutic portfolio for patients with this tumour (Figure 1).

## Challenges of precision therapeutic development

The design and conduct of biomarker-directed clinical trials that are adequately powered for small groups of patients carrying a diverse range of potentially actionable genetic aberrations, as in PDAC, is challenging, making the standard model of drug development inadequate (Biankin et al., 2015). To overcome these challenges, novel approaches using adaptive statistical designs and a master protocol to assign patients to different candidate drugs have been developed and shown promise in many tumour types (Hyman et al., 2017). In addition, recent efforts in molecular characterisation of PDAC from large-scale initiatives such as The Cancer Genome Atlas (TCGA), the International Cancer Genome Consortium (ICGC) and others have identified larger subgroup of PDAC patients based on genomic and transcriptomic changes who

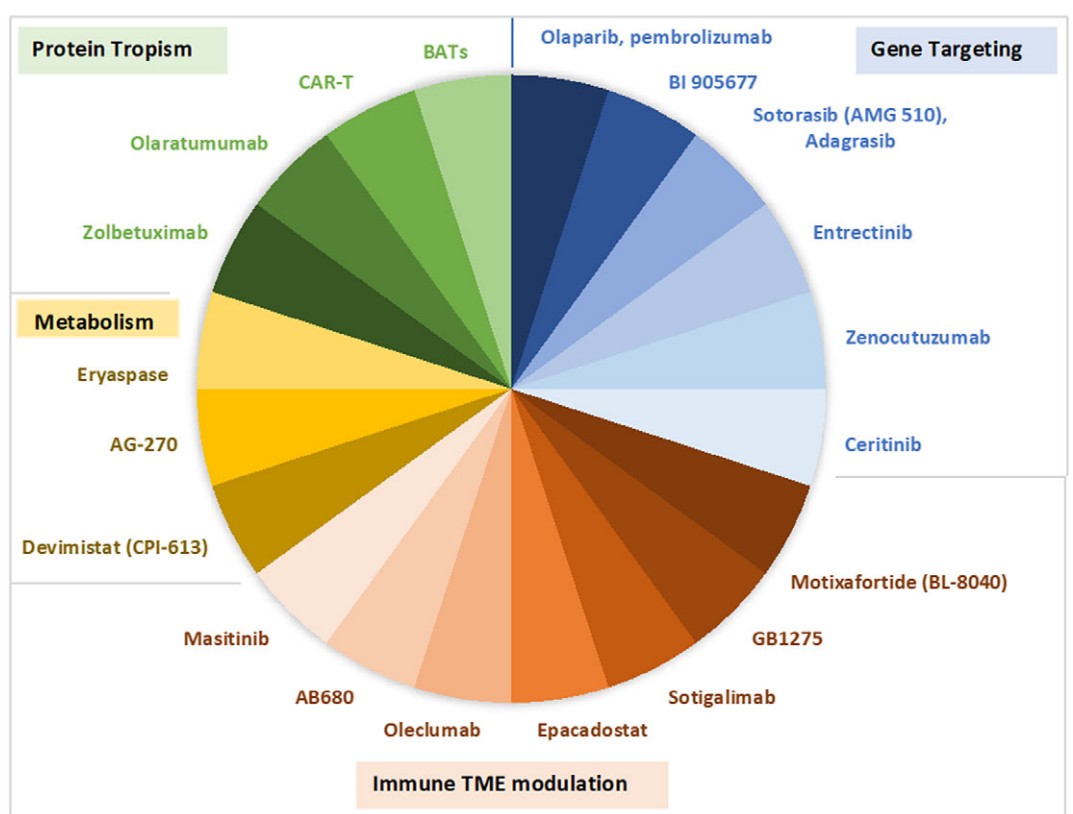

**Figure 1.** Selection of therapeutic targets and compounds under early clinical investigation in PDAC. TME, tumour microenvironment.

share molecular events beyond point mutations in coding genes and who are likely to benefit from certain treatments (Biankin et al., 2012; Chang et al., 2014; Waddell et al., 2015; Bailey et al., 2016; Collisson et al., 2019; Froeling et al., 2021). This approach is providing a unique opportunity to identify new therapeutic vulnerabilities in larger subgroups of patients (Chantrill et al., 2015; Beer et al., 2019, 2020; Dreyer et al., 2021, 2022; Froeling et al., 2021). Several clinical trials, including Precision-Promise, PASS-01, Precision-Panc, and others, are currently investigating novel precision therapeutic strategies based on a biomarker-enriched approach. Precision-Panc is a multi-centre national dynamic therapeutic development platform established in the UK in 2017 (Dreyer et al., 2019a,b; Froeling et al., 2021). Patient enrolled in the trial undergo endoscopic ultrasound or radiologically guided biopsy for suspected PDAC and subsequent integrated histology and molecular profiling assessing point mutations, copy number, structural variations, fusions and tumour mutational signatures (Dreyer et al., 2019a,b; Froeling et al., 2021). The results of molecular profiling obtained through the Precision-Panc Master Protocol may subsequently inform eligibility for enrolment in a PRIMUS study (Pancreatic canceR Individualised Multi-arm Umbrella Study), examining different biomarker-based treatment regimens (Froeling et al., 2021). In this way, by combining molecular information with clinical response data, and rapid forward and backward translation between the laboratory and the clinic, multiple hypothesis can be tested to investigate candidate biomarkers of therapeutic response and resistance. There are currently more than 300 participants enrolled in the Precision-Panc trials on offer, with additional studies anticipated to open (Froeling et al., 2021).

## Conclusions

PDAC is a devastating disease with limited therapeutic options (Figure 2). Despite major efforts in pre-clinical and clinical research, progress in PDAC lags behind other tumour types and the effects of the COVID-19 pandemic on this already critical situation are concerning (Casolino and Biankin, 2021; Casolino et al., 2021). In adjuvant setting, m-FOLFIRINOX is the international standard for non-Asian population. The combination of gemcitabine and capecitabine remains an option for patients considered unfit for m-FOLFIRINOX while single-agent gemcitabine is an option for patients who are not eligible for combination therapy. Primary systemic treatment is the standard of care for patients with borderline resectable and locally advanced disease while the role of NAT in resectable PDAC is still debated due to the lack of data from randomised trials. First-line FOLFIRINOX and gemcitabine/nab-paclitaxel can improve survival in patients with advanced disease. Nevertheless, survival remains poor and there is urgent need for new treatment strategies. While the identification of new therapeutic vulnerabilities is opening the door to novel treatment options, these are extremely rare in PDAC limiting the access to novel drugs to small patient subgroups and making the standard model of drug development challenging. In addition, molecular testing and targeted treatments are often not available in health care systems, particularly in lower-middle income countries, thus preventing the access to potentially active treatments for many patients. Continue investing in integrated preclinical and clinical research with a continuous forward, and backward analysis, is essential to identify new effective treatments and to implement novel models of biomarker-based clinical trials to improve the

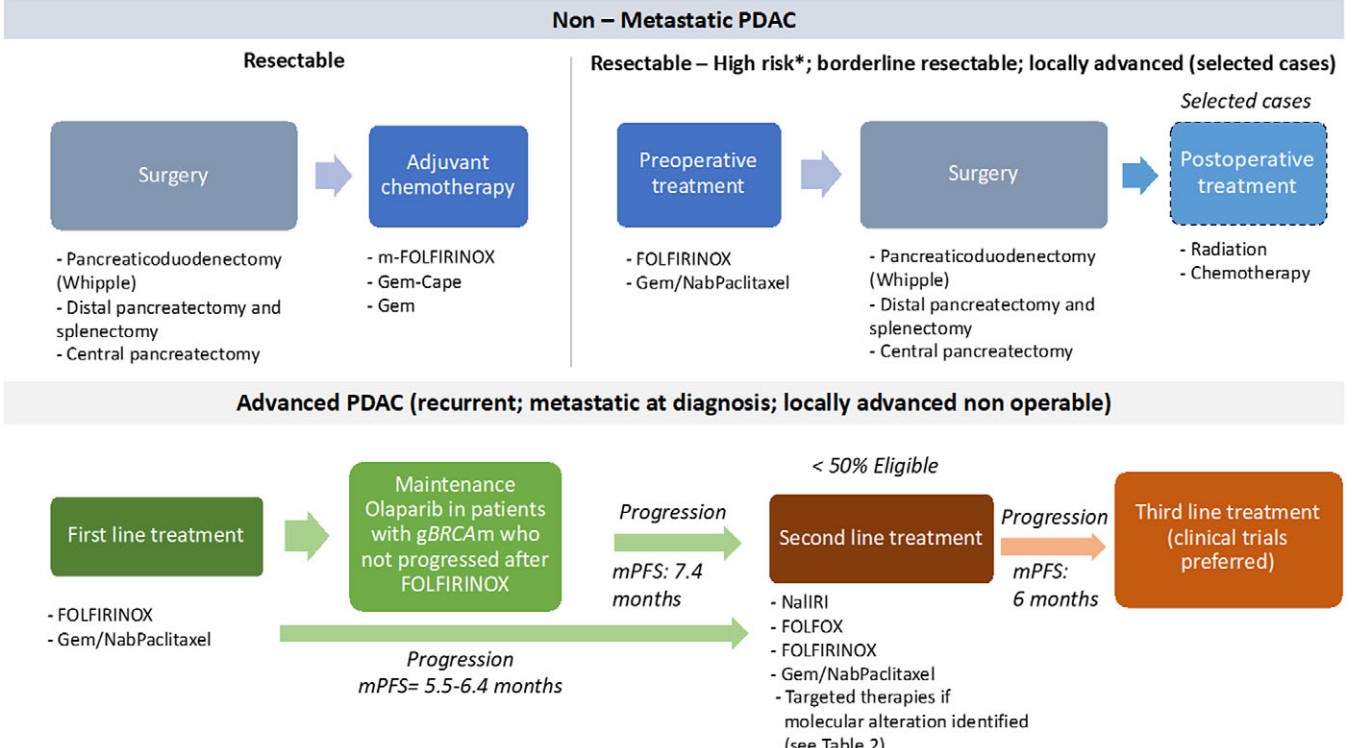

**Figure 2.** Overview of the standard treatment of PDAC.
*Note*: *High-risk patients are defined as follows: suspicious of advanced disease based on imaging findings or on significantly elevated CA19–9, large primary tumours or regional lymph nodes involvement, uncontrolled pain or excessive weight loss, high risk of pancreatic fistula. cape, capecitabine; gBRCAm, germline *BRCA* mutation; Gem, gemcitabine; Gem-Cape, gemcitabine plus capecitabine; mPFS, median progression-free survival. Enrollment in clinical trials should be always preferred, particularly after progression on second-line.

survival of patient with PDAC. In parallel, enhancing global oncology initiatives is warranted to bridging the gap between countries and ensures equitable access to the best and innovative care opportunities for every patient around the world.

**Open peer review.** To view the open peer review materials for this article, please visit http://doi.org/10.1017/pcm.2023.2.

**Author contributions.** Conceptualization: R.C., A.V.B.; Data curation: R.C., A.V.B.; Methodology: R.C., A.V.B.; Writing – original draft: R.C., A.V.B.; Writing – review and editing: R.C., A.V.B.

**Financial support.** This work was supported by Cancer Research UK C29717/A17263, C29717/A18484, C596/A18076, C596/A20921, A23526; Wellcome Trust Senior Investigator Award: 103721/Z/14/Z; Pancreatic Cancer UK Future Research Leaders Fund FLF2015_04_Glasgow; Scottish Genomes Partnership SEHHD-CSO 1175759/2158447; MRC/EPSRC Glasgow Molecular Pathology Node and The Howat Foundation. This funding source had no role in the design of this study and will not have any role during its execution, analyses, interpretation of the data or decision to submit results.

**Competing interest.** A.V.B.: BMS – personal and institutional financial interest; AstraZeneca – personal and institutional financial interest; Cumulus Oncology – Leadership role, stock ownership; Modulus Oncology – Leadership role, stock ownership; Wollemia Oncology – Leadership role, stock ownership; ConcR – Leadership role, stock ownership; Cambridge Cancer Genomics – Leadership role, stock ownership; Agilent Technologies – IP; financial interest; Novartis – personal and institutional financial interest; Gabriel Precision Oncology – Leadership role, stock ownership. R.C. declares no conflict of interest.

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
