## [Reviewer Report]

*Comments to Author*: This Review article summarizes in a comprehensive fashion the state-of-the-art regarding the therapeutic approaches to pancreatic ductal adenocarcinoma (PDAC). The authors nicely illustrate clear results obtained in the past years in the therapeutic approaches to PDAC as well as uncertainties which are still under debate or unclear.

Some language errors and difficult to read sentences are present throughout the text and the authors should re-check the all manuscript to smoothen these issues.

---

## [Reviewer Report]

*Comments to Author*: Thank you for giving me opportunity to read and review this comprehensive manuscript about the current treatment of PDAC. The whole work is written very synoptical and understandable and I believe it will find a wide range of readers seeking information about current medical treatment of PDAC with future perceptiveness.

Taking into a count authors include information about the latest experimental and scientific directions towards personalized medicine in the last part of their article I miss some information about the possibility of surgical treatment for metastatic disease in highly selected patients (DOI: 10.1111/ans.13738; DOI: 10.1016/j.ijsu.2017.10.066; etc….) despite we don’t have a randomized trials with high evidence on this topic.

---

## [Editor Report]

*Comments to Author*: Thank you for you submission. As indicated, the reviewers appreciated your thorough overview of current treatment strategies for PDAC patients, but the manuscript should be carefully read through to improve the clarity of some of the text and correct any confusing sentences.

---

## [Reviewer Report]

*Comments to Author*: Thank you for this revised version of this comprehensive article. Authors fullfiled all suggestions and I fully recomend it for publication in Precision Medicine.